# The Impact of Nordic Walking Pole Length on Gait Kinematic Parameters

**DOI:** 10.3390/jfmk8020050

**Published:** 2023-04-26

**Authors:** Luca Russo, Guido Belli, Andrea Di Blasio, Elena Lupu, Alin Larion, Francesco Fischetti, Eleonora Montagnani, Pierfrancesco Di Biase Arrivabene, Marco De Angelis

**Affiliations:** 1Department of Human Sciences, Università Telematica degli Studi IUL, 50122 Florence, Italy; 2Department of Sciences for Life Quality Studies, University of Bologna, 47921 Rimini, Italy; 3Department of Medicine and Aging Sciences, “G. d’Annunzio” University of Chieti-Pescara, 66100 Chieti, Italy; 4Department of Motor Activities, Petroleum Gas University Ploiesti, 100600 Ploiesti, Romania; 5Faculty of Physical Education and Sport, Ovidius University of Constanta, 900029 Constanta, Romania; 6Department of Basic Medical Sciences, Neuroscience and Sense Organs, University of Study of Bari, 70124 Bari, Italy; 7Department of Sports and Health Sciences, University of Brighton, Brighton BN2 4AT, UK; 8Department of Biotechnological and Applied Clinical Sciences, University of L’Aquila, 67100 L’Aquila, Italy

**Keywords:** Nordic walking, walking, 3D kinematics, biomechanics, gait analysis

## Abstract

Nordic walking (NW) is a popular physical activity used to manage chronic diseases and maintain overall health and fitness status. This study aimed to compare NW to ordinary walking (W) with regard to pole length and to identify kinematic differences associated with different poles’ length (55%, 65% and 75% of the subject’s height, respectively). Twelve male volunteers (21.1 ± 0.7 years; 1.74 ± 0.05 m; 68.9 ± 6.1 kg) were tested in four conditions (W, NW55, NW65 and NW75) at three different speeds (4-5-6 km∗h^−1^). Each subject performed a total of twelve tests in a random order. Three-dimensional kinematics of upper and lower body were measured for both W and NW, while oxygen consumption levels (VO_2_) and rating of perceived exertion (RPE) were measured only for NW trials with different poles’ length. NW showed a higher step length, lower elbow motion and higher trunk motion (*p* < 0.05) compared to W. Additionally, NW65 did not show any kinematic or RPE differences compared to NW55 and NW75. Only NW75 showed a higher elbow joint (*p* < 0.05) and lower pole (*p* < 0.05) range of motion compared to NW55 and a higher VO_2_ (*p* < 0.05) compared to NW55 and NW65 at 6 km∗h^−1^. In conclusion, the use of the poles affects the motion of the upper and lower body during gait. Poles with shorter or longer length do not produce particular changes in NW kinematics. However, increasing the length of the pole can be a smart variation in NW to increase exercise metabolic demand without significantly affecting the kinematics and the RPE.

## 1. Introduction

Walking with poles in an urban or natural environment is commonly known as Nordic walking (NW), defined as an outdoor non-competitive fitness activity in which the practitioners perform brisk walking with specially designed poles, similar to those for cross country skiing, engaging also the upper body [1,2,3]. NW is widely and successfully applied in the management of several not-transmissible chronic diseases, such as diabetes, cancer, hypertension, obesity [4,5,6,7,8,9] or in case of joints diseases [10,11] or for general health maintaining purpose [12,13].

Over an approximate period of 25 years, NW has received great attention by researchers. The main attractive feature of NW, as a fitness activity, is the suggestion from the literature about a higher request in terms of energy expenditure with respect to ordinary walking (W). The average rate of oxygen uptake (VO_2_) is reported to be significantly higher in NW with respect to W at the same speed [3,14,15,16,17,18,19,20,21], without any increase in the rate of perceived exertion [14,17,19,20,21]. These results reinforce the practice of NW as an activity for weight loss and maintaining an optimal health status. 

From a biomechanical point of view, NW is often compared to W with the aim to understand if the use of the poles could affect the joint load. For this reason, the lower limbs are mainly investigated [22], in particular the knee joint, obtaining, however, controversial results [2,23,24,25,26] due to the different tested samples in which not always the knee joint load was reduced in NW with respect to W. The upper body is mainly studied in terms of muscular activation patterns [27]. Some studies suggest that NW can lead to increased shoulder mobility and reduced tenderness in the shoulder girdle [28], as well as decreased neck and shoulder pain [29]. In general, NW increases the walking distance and speed, as well as the stride length, decreasing the cadence [30]. Moreover, NW increases the muscle activation and the strength of upper limbs, as well as the upper and lower range of motion and the ground reaction force [30].

According to the actual literature state of art, NW is deeply investigated but less attention is given to the main element that differentiates NW from W: the pole. In NW, the poles are investigated most of all for the force transmission profile [31,32] and it is well-known that properly pushing the pole on the ground allows to increase the NW metabolic benefits. Not by chance, there is a difference in the applied force on the ground between NW instructors and recreational individuals [32]. Despite that, at present, few information is available about the poles’ length and how this parameter can affect the body kinematics. Usually, the NW poles are set approximately at 2/3 of the subject’s height (height of the subject in centimeters∗0.68) [23,32,33], but it is common to use them based on comfort perception. For this reason, it is important to understand the number of kinematic modifications that different pole lengths could induce. From a metabolic perspective, some results are available about both different weight and length of the poles. About weight, no differences were measured comparing NW with heavier (adding 0.5; 1 and 1.5 kg to the pole) or standard poles on metabolic data, therefore there is no suggestion to use heavier poles to increase the metabolic demand [34]. About length, poles lengths which are self-selected caused greater energy expenditure than shorter poles only in uphill walking but not on level walking [3]. No information is present about using longer poles and no information is available about the kinematics of NW with different poles’ length. Moreover, the increased metabolic demand of NW compared to W is well-known, but less is known about the relationships between kinematics and VO_2_ consumption during NW exercise.

It is reasonable to hypothesize that modifying the length of NW poles could affect gait kinematics, as well as perceived fatigue and metabolic demand during exercise. Modifying the poles’ length could lead to using the upper limbs in a different way to properly manage the pole, and this modification could affect the motion of the spine and lower limbs. Thus, this study aimed to (1) characterize the kinematics of NW compared to regular walking, with a focus on upper body motion and (2) examine the effects of different pole lengths on kinematics, metabolic demand and perceived fatigue during NW exercise. The primary practical goal of this research is to provide evidence-based recommendations for practitioners and instructors involved in NW training.

## 2. Materials and Methods

### 2.1. Design and Participants

This is a cross-sectional study design with a random crossover approach. Participants were recruited as part of the undergraduate program of Sport Science at the University of L’Aquila. The participation was on a voluntary base and the specific criteria of selection included: no history of musculoskeletal or neurological pain in the last 6 months and to be physically active. Participants never practiced NW. This was required as we wanted to resemble as close as possible the activity of recreational and novice NW practitioners; expert participants would have displayed different walking techniques and styles, biasing the results. Due to the inexperience of the participants, they attended three preliminary weeks of supervised training with a NW instructor, to avoid a learning effect during the study as Figard-Fabre et al. found [21]. The instructor taught the basic technique (diagonal technique) according to the INWA guidelines [35]. Diagonal technique can be briefly described as technique that seeks a position of the trunk inclined forward, with long strides and active use of the arms without any limitation of flexion or extension [36]. Practices were performed using a treadmill, at the same time of the day, 3 sessions/week, 1 h/session. 

A total amount of 20 male participants were recruited but only 12 completed the experiment (mean ± SD: 21.1 ± 0.7 years; 1.74 ± 0.05 m; 68.9 ± 6.1 kg; 22.7 ± 1.6 BMI; 12.8 ± 2.7% body fat). A power analysis of the sample was conducted, and 12 subjects were sufficient to satisfy a power at 80% and an error probability at 5%. During the recruiting phase and prior to the training phase, all participants were informed and gave voluntary consent to participate in the study, and privacy criteria were meet. The study was approved by the Ovidius University of Constanta Nr. 141 din 21 February 2023 in accordance with the Declaration of Helsinki.

### 2.2. Instrumentation

The NW pole used was customized with a telescopic pole (Skitrab, Bormio, Italy; 210 g of mass each) connected with a grip and a tip specific for NW (Swix, Lillehammer, Norway). The choice to use a telescopic pole allowed to modify the length but not the weight of the pole for each trial.

Kinematic data were collected using a 4 infrared 3-dimensional camera SMART integrated System working at a sample rate of 60 Hz (BTS Bioengineering, Garbagnate Milanese, Italy.

Metabolic data were collected using a portable metabolimeter K4b2 (Cosmed, Roma, Italy), calibrated before each test session.

All the tests were performed on a Cosmed T170 treadmill (Cosmed, Roma, Italy), adapted by taking out frontal and lateral handlebars to allow participants to perform the typical NW gesture without hitting the structure. The treadmill was calibrated according to the literature [37] before the beginning of the experimental procedures.

The Figure 1 resumes the used instrumentation.

### 2.3. Procedure and Data Collection

Testing procedures were carried out in a Sport Performance Laboratory at a mean temperature of 19 °C and a mean relative humidity of 51%, and each subject was tested at the same time of the day, avoiding any circadian effect [38,39,40]. The first day of the training period, all participants were tested for anthropometric measures aiming to find out the three different poles’ length: classical, reduced and increased length 65%, 55% and 75% of the participant’s height, respectively. The supervised training period was performed with classical length poles (65% of subject’s height) as well as longer and shorter poles in a random order. The last day of training, each individual received a personalized random sequence of the test conditions.

Each participant performed a total amount of 12 tests. The experimental procedure had 4 conditions, each condition was tested during a single day and each test day was separated by 7 days or rest. The test conditions were: ordinary walking (W) and NW with poles at different lengths (65%, 55%, 75% of the subject’s height defined as NW65, NW55, NW75). The length of the pole was measured from the insertion of the lace in the handle to the tip final extremity (Figure 2). Each condition was tested at 3 different fixed speeds 4-5-6 km∗h^−1^ [41] and no slope was applied to the treadmill. Conditions were randomly assigned while speeds were always administered from the slowest to the faster one. Each trial lasted 10 min and a resting period of 20 min was observed between each trial test. Table 1 resumes a real example of randomized sequence for one single participant. During tests, kinematic data, rating of perceived exertion (RPE) and oxygen consumption (VO_2_) were collected.

#### 2.3.1. Kinematic Data

Kinematic analysis was conducted using a model of 18 body reflective markers (Figure 3), positioned on the most used body landmarks according to other studies about NW or motion analysis [2,23,25,26,42,43,44,45,46,47]. 

The last two minutes of each trial were taken into account for the kinematic analysis, aiming to reduce the stride variability [48]. Smoothing procedures and missing frames were managed through the SMART Analyzer software (BTS Bioengineering, Garbagnate Milanese—Italy). Once the raw kinematic data were processed, a ratio between the step length and the elbow horizontal displacement was also calculated (SL/EHD ratio), with the aim to understand if the use of the poles addressed some changes in the upper limbs motion in NW with respect to W. All the kinematic data were measured on the right side of each subject in order to standardize the process. Table 2 resumes and describes the kinematic parameters acquired for each test.

#### 2.3.2. Fatigue and Metabolic Data

VO_2_ was measured using a wearable metabolimeter during the whole duration of each test and the last seven minutes were taken into account for the metabolic analysis. The evaluation of the average oxygen consumption of each NW trial was used to look for correlations between kinematics and metabolic demand in NW exercise. Due to the well-known metabolic difference between NW and W [3,14,15,16,17,18,19,20,21], VO_2_ consumption during W was not considered in this study. Anyway, the metabolimeter was worn as well even during W trials in order to avoid perceived comfort difference with respect to NW. Immediately after each trial, the participants indicated their rating of perceived exertion using the category rating-10 (CR-10) scale modified by Foster et al. [49].

### 2.4. Statistical Analysis

All data were tested with the Shapiro–Wilk’s test for normality. As data were normally distributed, parametric inferences were used for the analysis. Kinematic differences between NW65 and W were tested using the Student paired *t*-test, while to measure kinematics differences in NW with different poles’ height, an ANOVA design for repeated measures with Sidak correction was used. Finally, the Pearson correlation coefficient was used to look for a correlation between kinematic and metabolic data. The significant level was set at *p* = 0.05 and the data were analyzed using SPSS (SPSS Inc., Chicago, IL, USA).

## 3. Results

### 3.1. Kinematic Differecences between NW65 and W

According to statistical analysis, kinematic data in NW65 showed significant differences with respect to W, for all the three tested speeds. Most of the significant differences involved the upper body segments (Table 3).

The use of the poles significantly increased both the C7 and the S1 vertical displacement for all speeds; even the elbow horizontal displacement and the elbow advancing speed resulted significantly lower in NW compared to W, while the step length was higher in NW in comparison with W for all tested speeds.

### 3.2. Kinematic Differecences between NW with Different Poles’ Length

A limited number of significant differences were found for kinematics between NW with different poles’ length (Figure 4). For all the three tested speeds, no differences were measured between NW65 and NW75, while only one significant difference was measured at 6 km∗h^−1^ between NW55 and NW65 for the pole Δ slope (31.7 ± 2.6° and 28.0 ± 3.2°, respectively, *p* = 0.013). Significant differences were measured between the shorter and the longer poles’ length, NW55 and NW75, particularly on the elbow and poles kinematics. At 4 km∗h^−1^, three significant differences were measured between NW55 and NW75: (1) pole minimum slope (14.5 ± 6.6° and 25.7 ± 10.7°, respectively, *p* = 0.013); (2) pole maximum slope (41.8 ± 4.4° and 49.5 ± 4.0°, respectively, *p* = 0.028); (3) pole Δ slope (27.3 ± 3.2° and 23.4 ± 2.5°, respectively, *p* = 0.010). At 5 km∗h^−1^, four significant differences were measured between NW55 and NW75: (1) elbow Δ angle (30.7 ± 6.5° and 41.9 ± 13.3°, respectively, *p* = 0.027); (2) pole minimum slope (13.5 ± 5.5° and 27.4 ± 9.0°, respectively, *p* = 0.000); (3) pole maximum slope (42.9 ± 4.0° and 51.1 ± 7.1°, respectively, *p* = 0.001); (4) pole Δ slope (29.4 ± 2.7° and 24.7 ± 2.8°, respectively, *p* = 0.003). Finally, at 6 km∗h^−1^, four significant differences were measured between NW55 and NW75: (1) elbow Δ angle (31.7 ± 7.1° and 42.8 ± 12.0°, respectively, *p* = 0.031); (2) pole minimum slope (13.7 ± 6.4° and 27.7 ± 7.0°, respectively, *p* = 0.000); (3) pole maximum slope (45.5 ± 4.3° and 54.2 ± 5.3°, respectively, *p* = 0.002); (4) pole Δ slope (31.7 ± 2.6° and 26.5 ± 3.0°, respectively, *p* = 0.000). The results for all the data measured can be read in Appendix A.

### 3.3. Correlations between Metabolic Data and Kinematics in NW

No differences were measured in RPE for NW with different poles’ lengths at the same speed; while two significant differences were measured at 6 km∗h^−1^ between NW55 and NW75, as well as between NW65 and NW75 for VO_2_ (21.7 ± 1.7 mL∗kg^−1^∗min^−1^ and 24.1 ± 2.2 mL∗kg^−1^∗min^−1^, respectively, *p* = 0.002 and 22.2 ± 2.9 mL∗kg^−1^∗min^−1^ and 24.1 ± 2.2 mL∗kg^−1^∗min^−1^, respectively, *p* = 0.030) (Figure 5).

Significant correlations were measured between metabolic data and kinematics in NW. In NW65 at 6 km∗h^−1^, the S1 vertical displacement significantly correlated with VO_2_ (r = 0.59 *p* = 0.043). In NW75 at 4, 5 and 6 km∗h^−1^, significant correlations were found between the spine forward slope and the VO_2_ (r = 0.71 *p* = 0.010; r = 0.69 *p* = 0.014; r = 0.71 *p* = 0.010, respectively, for each tested speed). In NW75, another two correlations were measured: one at 4 km∗h^−1^ between poles Δ slope and VO_2_ (r = 0.63 *p* = 0.027) and another one at 6 km∗h^−1^ between elbow Δ angle and VO_2_ (r = 0.61 *p* = 0.036). No significant correlations were found at any speed for NW55.

## 4. Discussion

This study aimed to characterize the kinematics of NW compared to W and to examine the effects of different pole lengths on kinematics, metabolic demand and perceived fatigue during NW. The study of the effect of different poles’ length on gait kinematics is the main novelty of the investigation. Additionally, exploring the relationship between NW kinematics and exercise VO_2_ consumption could provide a new perspective on NW training and practice.

The main attention in NW from a biomechanical point of view is usually focused on lower limbs [22]. Conversely, the most interesting aspect of this investigation is the use of a 3D kinematic model for the study of the upper body segments. Finally, by utilizing a treadmill, we were able to maintain a constant speed throughout the tests, resulting in a more accurate comparison of the impact of various pole lengths on both kinematic and metabolic data.

### 4.1. Kinematic Differecences between NW65 and W

Significant differences in kinematics were found for all the tested speeds between NW65 and W. The use of the poles during walking seems to affect the gait kinematics both for the upper and lower body. The measured differences are significant but relatively small. This point can be considered delusive, but it should be considered that the NW is a physical activity that is performed for a long time during the training session, therefore it is logical to mind that even small differences can affect the final energy expenditure of the exercise.

The poles affect the vertical displacement of the two segments of the spine, C7 and S1, increasing the muscular work against gravity. In fact, in NW65, a moderate positive but significant correlation is present between the vertical displacement of S1 and the VO_2_. The use of the poles in NW not only allows a higher use of the upper limbs musculature [50] but engages all the trunk muscles [51,52,53], reflecting a higher vertical displacement of the whole spine.

With respect to the upper limb motion, particular attention was paid to the elbow. A significant difference for the angle displacement of the elbow was measured between NW and W only for the slower speed, 4 km∗h^−1^, in fact NW showed a higher angle displacement with respect to W. This difference disappeared when the gait speed increases, higher speeds showed a similar elbow angle. On the other hand, significant differences were observed between NW and W at all tested speeds, both in terms of elbow horizontal displacement and elbow advancing speed. In both cases, the NW values were smaller with respect to W, this can be addressed by the presence of the poles. It is reasonable to hypothesize that the pole’s presence can affect the motion of the upper limbs both on the coordinative and on the kinematics profile, due to the necessity to push the pole against the ground [32]. It is well-known that the pole’s weight does not affect the oxygen consumption, but only the muscle activation [34]; thus, the motion of the elbow is mainly affected by the pole’s usage but not by its weight. The result of the present study cannot confirm at all this hypothesis but the presence of the positive significant correlation for NW75 between the elbow angle displacement and the VO_2_ can suggest that more research in this area is needed to clear this aspect. 

For the lower body kinematics, NW showed a significantly higher step length with respect to W and this aspect could explain the advantages of NW for special populations, such as diabetic people [54,55]. To increase the step length results in engaging the ankle and hip motion much more, therefore this information can be relevant in all cases where higher ankle and hip motion is required: such as sedentary or pathological individuals [10,11,56]. The results of this research, about step length and step frequency, are consistent with previous studies [19,25] but with different absolute values. Willson et al. [25] observed an increase in step length up to 20 cm, whereas the difference observed in the present research is smaller (approximately 3 cm). However, it is important to note that there are methodological differences between the two experimental protocols as other researchers used self-selected and faster walking speeds compared to the present study. Therefore, it is reasonable that the difference in step length could depend on the absolute different speed used for the tests. 

Finally, as a direct consequence of the effect of the poles on the elbow motion and step length, even the SL/EHD ratio is significantly different between NW and W. The SL/EHD ratio is the mathematical ratio between the step length and the elbow horizontal displacement. It could be considered as a coordinative aspect of the gait. For all tested speeds, NW showed higher values of this ratio compared to W, because NW led to higher step length but smaller elbow horizontal displacement. This information confirms that the use of the poles can modify the gait spatiotemporal parameters.

### 4.2. Characterization of NW with Different Poles’ Length

The second aim of this research was to examine the effects of different pole lengths on kinematics, metabolic demand and perceived fatigue during NW. Usually, the suggested length of the poles is set approximately at 2/3 of the subject’s height [23,32,33]; thus, in this research, the poles were set at 55%, 65% and 75% of the subject’s height. It would have been plausible to hypothesize that the use of shorter or longer poles could have influenced the spine inclination or the upper limb motion with respect to the classical length, but no statistical kinematic differences were measured between NW65 and NW55 or NW75 for all the tested speeds. No statistical differences were measured even for exercise RPE, suggesting that the length of the poles does not affect the perceived fatigue at any speed. 

The use of different poles’ length determined a limited number of significant differences both on the kinematic and metabolic profile: all the differences were related to the NW75 condition. In fact, NW75 showed a higher elbow angular displacement and a lower change of poles’ slope with respect to the ground compared to NW55. The significant difference was present for all tested speeds. It is rationale to assume that the higher elbow angular displacement and the position of the pole constantly inclined with respect to the ground enable the participants to optimize the use of the pole during the pushing phase. The latter interpretation can be read in light of the significant increase of VO_2_ consumption at 6 km∗h^−1^ measured in NW75 compared to NW55 and NW65. These results are consistent with literature that suggests keeping the pole inclined for pushing optimization [21]. Therefore, the use of a longer pole can be beneficial for NW practitioners due to the increased metabolic demand without any effect on kinematics and exercise RPE compared to classical length. On the other hand, no reasons seem to be present for suggesting to reduce the length of the pole (NW55) in level walking, whereas it seems useful on uphill walking [3]. 

The absence of substantial significant kinematic differences between the different pole length conditions may seem odd, but it is important to consider that the sample for this research was deliberately chosen to have no previous experience. Although the participants underwent a training period before the tests, they were not as skilled as regular practitioners. Therefore, it is plausible to assume that the absence of specific NW coordination and technique allowed the participants to respond in a similar way to the different pole lengths. This perspective can be viewed positively when working with novice subjects, but this data cannot be applied to skilled practitioners without specific research.

Based on the research findings, NW instructors can opt for a different pole length compared to the traditional one when instructing novices. This is because it is widely known that the level of comfort does not vary between poles with different lengths [3]. Furthermore, novice practitioners may be advised by NW instructors to use a longer pole length (75% of the subject’s height) if their objective is to increase metabolic demand during exercise without altering the kinematics of the movement and the exercise RPE. This recommendation may be particularly useful in the management of metabolic clinical conditions, such as overweight and obesity [55,57,58]. This practical application can be considered the main novelty of this research.

### 4.3. Limitations

The findings of this study are applicable solely to young male novice NW practitioners. It would be highly interesting to replicate the same experimental protocol using participants of different genders and ages. Additionally, this research did not provide any insights into the subject’s adaptation or learning process during the training period, nor did it examine the comfort levels associated with the use of a specific pole during extended training sessions. Therefore, this study may be viewed as a preliminary exploration of the topic, and further research is warranted to enhance the practical information available to NW instructors.

## 5. Conclusions

Nordic walking (NW) differs from ordinary walking in both kinematic and metabolic profiles. The use of poles influences the movement of the upper and lower body. The length of the poles, whether shorter or longer than the standard size (about 2/3 of the subject’s height), does not significantly alter the kinematics of NW, with the exception of a comparison between longer and shorter poles, where longer poles significantly impact elbow angular displacement and pole angle relative to the ground. These differences may explain the higher VO_2_ consumption associated with longer poles, without any corresponding increase in exercise RPE. In summary, this study sheds light on pole-length selection, suggesting that reducing the length of poles offers no benefit to ground-level NW, while increasing pole length provides a viable method to increase metabolic demand without impacting kinematics or perceived fatigue. Furthermore, the use of telescoping poles may offer a convenient way to adapt poles to varying needs.

## Figures and Tables

**Figure 1 jfmk-08-00050-f001:**
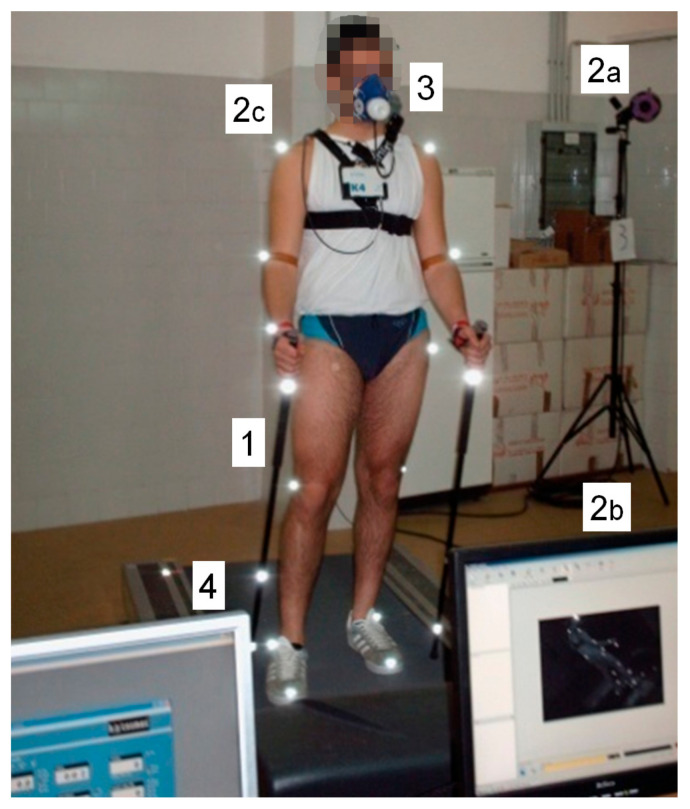
Schematic representation of the experimental setting. 1—Nordic walking pole. 2—3D kinematic integrated system; 2a—Infrared camera; 2b—Reflective marker; 2c—Workstation. 3—Metabolimeter. 4—Treadmill and workstation for treadmill management.

**Figure 2 jfmk-08-00050-f002:**
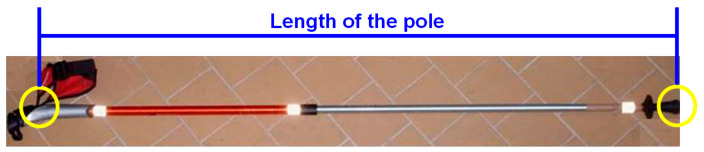
Landmarks for measurement of NW pole’s length, from left to right: insertion of the lace in the handle and tip final extremity.

**Figure 3 jfmk-08-00050-f003:**
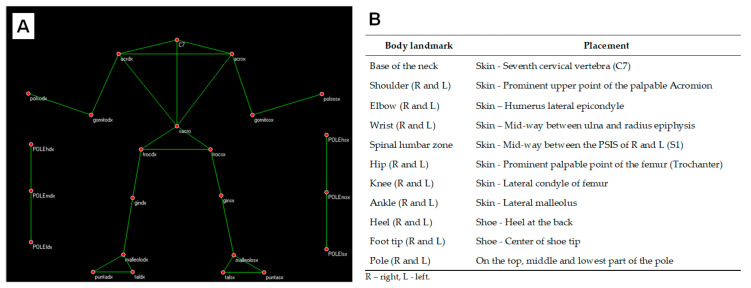
Position of the reflective markers. (**A**)—Kinematic used model for the automatic recognition of the body landmarks. (**B**)—Detailed description of the position for each reflective marker on body landmarks.

**Figure 4 jfmk-08-00050-f004:**
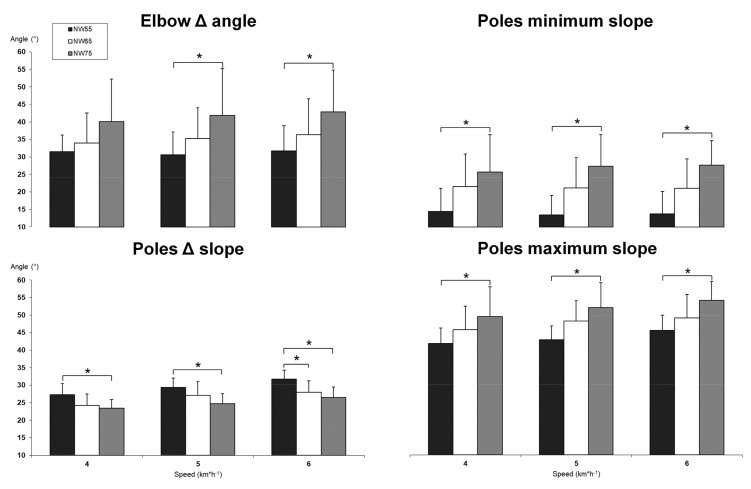
Kinematic differences between NW using different poles’ length. NW55—Nordic walking with pole length adjusted at 55% of subject’s height. NW65—Nordic walking with pole length adjusted at 65% of subject’s height. NW75—Nordic walking with pole length adjusted at 75% of subject’s height. * significant difference.

**Figure 5 jfmk-08-00050-f005:**
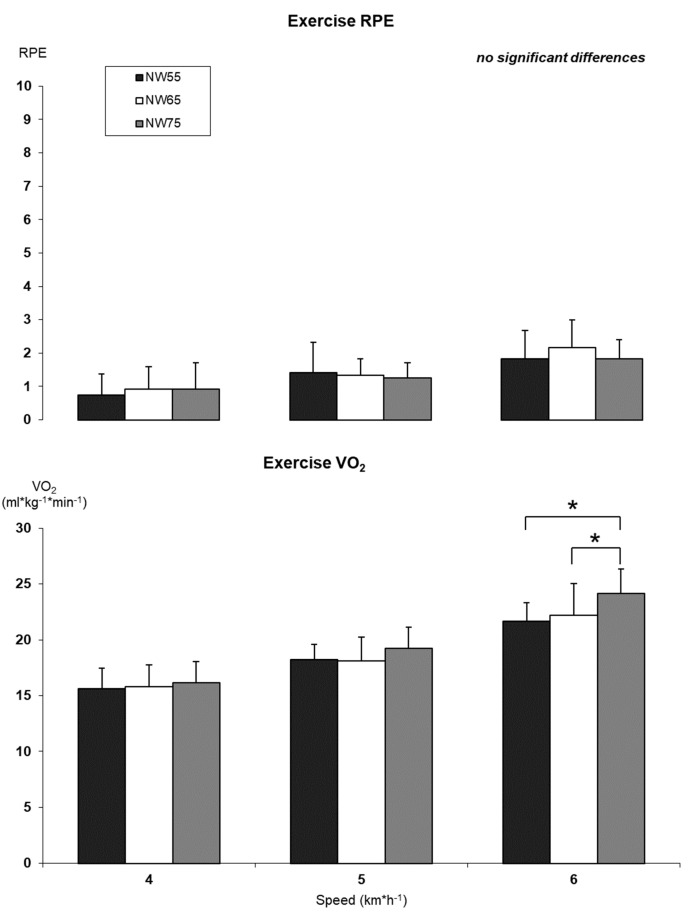
Fatigue perception and oxygen consumption differences between NW using different poles’ length. RPE—Rating of perceived exertion. VO_2_—Oxygen consumption. NW55—Nordic walking with pole length adjusted at 55% of subject’s height. NW65—Nordic walking with pole length adjusted at 65% of subject’s height. NW75—Nordic walking with pole length adjusted at 75% of subject’s height. * significant difference.

**Table 1 jfmk-08-00050-t001:** Real tests sequence of a participant. The sequence of the conditions was randomly assigned while the speeds were always administered from the slowest to the faster one (4-5-6 km∗h^−1^). Each trial, represented as an “X”, lasted 10 min and a resting period of 20 min was observed between each trial test.

Condition	Week 1			Week 2			Week 3			Week 4		
	Speed (km∗h^−1^)			Speed (km∗h^−1^)			Speed (km∗h^−1^)			Speed (km∗h^−1^)		
	4	5	6	4	5	6	4	5	6	4	5	6
W							X	X	X			
NW55	X	X	X									
NW65										X	X	X
NW75				X	X	X						

W—Walking. NW55—Nordic walking with pole length adjusted at 55% of subject’s height. NW65—Nordic walking with pole length adjusted at 65% of subject’s height. NW75—Nordic walking with pole length adjusted at 75% of subject’s height.

**Table 2 jfmk-08-00050-t002:** Detailed description of each parameter measured with kinematic analysis.

	Parameter	Description
Upper body	C7 vertical acceleration peak (m∗s^−2^)	Peak vertical acceleration after the push off phase of the foot on the ground
	C7 vertical displacement (cm)	Vertical displacement of the marker on C7 from the lowest to the highest point for each step
	Elbow horizontal displacement (cm)	Maximal horizontal displacement of the elbow on the sagittal plane for each step
	Elbow Δ angle (°)	Angular displacement of the elbow from the maximum flexion to the maximum extended position
	Elbow advancing speed (m∗s^−1^)	Speed of the elbow during the forward movement from back to front
	Spine forward slope (°)	Slope of the spine calculated by the inclination of the S1-C7 segment from the vertical position
	S1 vertical displacement (cm)	Vertical displacement of the marker on S1 from the lowest to the highest point for each step
Lower body	Step length (m)	Horizontal displacement of feet
	Step frequency (Hz)	Number of steps per second
Poles	Tip sagittal displacement (m)	Horizontal displacement of the pole tip
	Movement frequency (Hz)	Number of pole pushes per second
	Minimum slope (°)	Minimum inclination of the pole
	Maximum slope (°)	Maximum inclination of the pole
	Δ Slope (°)	Angular displacement of the pole from the maximum to the minimum inclination of the pole

Measures for elbows, feet and poles were taken all on the right side for the entire sample. Note: all the data are expressed as an average value of all gait cycles.

**Table 3 jfmk-08-00050-t003:** Differences between NW65 and W for kinematics.

Parameter	4 km∗h^−1^			5 km∗h^−1^			6 km∗h^−1^		
	NW Mean (SD)	W Mean (SD)	*p* Value	NW Mean (SD)	W Mean (SD)	*p* Value	NW Mean (SD)	W Mean (SD)	*p* Value
C7 vertical acceleration peak (m∗s^−2^)	3.0 (0.3)	3.0 (0.3)	0.805	3.3 (0.3)	3.1 (0.3)	0.127	3.7 (0.4)	3.5 (0.4)	0.002 *
C7 vertical displacement (cm)	3.5 (0.5)	3.2 (0.5)	0.010 *	4.7 (0.7)	4.0 (0.7)	0.000 *	5.7 (1.0)	5.1 (0.8)	0.007 *
Elbow horizontal displacement (cm)	10.9 (4.3)	15.9 (2.9)	0.002 *	12.7 (3.4)	17.4 (2.9)	0.001 *	14.1 (4.3)	18.3 (3.3)	0.001 *
Elbow Δ angle (°)	34.0 (8.6)	22.1 (6.5)	0.005 *	35.3 (8.8)	29.8 (5.7)	0.141	36.4 (10.3)	36.5 (6.9)	0.970
Elbow advancing speed (m∗s^−1^)	0.2 (0.1)	0.3 (0.1)	0.003 *	0.2 (0.1)	0.3 (0.1)	0.001 *	0.3 (0.1)	0.4 (0.1)	0.000 *
Spine forward slope (°)	9.3 (2.6)	8.7 (2.1)	0.109	10.0 (3.3)	9.1 (2.3)	0.032 *	10.9 (2.5)	10.1 (2.1)	0.067
S1 vertical displacement (cm)	3.1 (0.6)	2.7 (0.6)	0.001 *	4.5 (1.5)	3.7 (0.8)	0.033 *	5.5 (1.1)	4.9 (0.9)	0.020 *
Step length (m)	0.66 (0.04)	0.64 (0.03)	0.036 *	0.74 (0.04)	0.71 (0.04)	0.000 *	0.80 (0.05)	0.77 (0.04)	0.006 *
Step frequency (Hz)	0.85 (0.08)	0.86 (0.03)	0.750	0.91 (0.04)	0.95 (0.03)	0.004 *	0.99 (0.05)	1.02 (0.03)	0.010 *
SL/EHD ratio (cm)	7.3 (4.1)	4.12 (0.8)	0.023 *	6.3 (2.0)	4.2 (0.7)	0.004 *	6.4 (2.5)	4.4 (0.8)	0.005 *

W—walking. NW65—Nordic walking with pole length adjusted at 65% of subject’s height. SL/EHD ratio—ratio between the step length and the elbow horizontal displacement. * significant difference between NW and W.

## Data Availability

The data that support the findings of this study are available from the corresponding author, upon reasonable request.

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
