# Peer review of "The Impact of Nordic Walking Pole Length on Gait Kinematic Parameters"

_jfmk, 2023, doi:10.3390/jfmk8020050_

Round 1
Reviewer 1 Report
Introduction
1. In line No. 47……..Please don’t write exactly 25 years…..instead it can be written as ‘over an approximate period of 25 years’
2. Please simplify the sentence spanning line No. 47 to 49
Methods
1. In line No. 135………Why it is specifically for ‘The supervised training period was performed with 135 classical length poles (65% of subject’s height)’…what about the training and conditioning with reduced and increased pole height?
2. In line No. 144……..Any justification on the selection of different speed categories through the available UpToDate literature on the said topic?
3. In line No. 183….Authors may consider the relocation of the table 2 as a supplement material as it is the OEM (BTS Inc.) specific standard operating procedure on the placement of the markers.
4. In line No. 202…….Authors mentioned about ‘All data were tested with the Kolmogorov-Smirnov test for normality’….However, it is necessary to know that for a sample size <50, the Shapiro-Wilk’s test is more suitable to conduct the test for normality. Therefore, authors are requested to conduct the Shapiro-Wilk’s test and report accordingly.
Discussion:
1. In line No. 297……..Please replace the ‘extremities’ with ‘segment’
2. In line No. 343…….. Please replace the ‘eight’ with ‘height’
Limitations and Conclusions look reasonable without further modification.
Author Response
Dear Editor and Reviewer,
We would like to thank you for the time allowed to this review process. As a result, we are submitting the revised version for a possible publication in this respectable Journal. Below, you can find our responses; each comment is followed by its respective reply. We made changes in the manuscript in order to address suggestions and make it clearer for the readers, we underlined in yellow the responses to your comments and we used the track changes to correct some misprint or to enhance some phrases of the manuscript. All authors have made sufficient contributions and have approved the submitted manuscript.
Sincerely,
The Authors
Legend:
R1(Reviewer 1)
A (Authors)
1) R1:
In line No. 47……..Please don’t write exactly 25 years…..instead it can be written as ‘over an approximate period of 25 years’.
A:
Than you for the advice. Done. Line 48
2) R1:
Please simplify the sentence spanning line No. 47 to 49.
A:
Done. We divided in two sentences.
3) R1:
In line No. 135………Why it is specifically for ‘The supervised training period was performed with 135 classical length poles (65% of subject’s height)’…what about the training and conditioning with reduced and increased pole height?
A:
Thanks for the question. It is a typo of the final draft. The whole sentence was “The supervised training period was performed with classical length poles (65% of subject’s height) as well as longer and shorter poles in a random order.” Thanks for the advice, we checked the previous version of the paper and we correted. Lines 149-150.
4) R1:
In line No. 144……..Any justification on the selection of different speed categories through the available UpToDate literature on the said topic?
A:
Done, reference added. Line 158.
5) R1:
In line No. 183….Authors may consider the relocation of the table 2 as a supplement material as it is the OEM (BTS Inc.) specific standard operating procedure on the placement of the markers.
A:
Thanks for the comment. Because your advice was very similar to the one of Reviewer 2 about a better relocation of the table two we accepted the suggestion of Reviewer 2 and we create a single Figure with Figure 2 and Table 2. Line 183-187.
6) R1:
In line No. 202…….Authors mentioned about ‘All data were tested with the Kolmogorov-Smirnov test for normality’….However, it is necessary to know that for a sample size <50, the Shapiro-Wilk’s test is more suitable to conduct the test for normality. Therefore, authors are requested to conduct the Shapiro-Wilk’s test and report accordingly.
A:
Thanks for the suggestion. We perfomed the Shapiro-Wilk’s test and the normality was assessed as well. We corrected the sentence. Line 217.
7) R1:
In line No. 297……..Please replace the ‘extremities’ with ‘segment’.
A:
Thank you. Done. Line 311.
8) R1:
In line No. 343…….. Please replace the ‘eight’ with ‘height’.
A:
Thank you. Done. Line 357.
9) R1:
Limitations and Conclusions look reasonable without further modification.
A:
Thank you very much, we appreciated.
Reviewer 2 Report
I was asked to review the paper 3D Kinematic Analysis of Walking and Nordic Walking with 2 Different Poles’ Length. The work is very nicely written. I have quite a few comments, but they are of the clarifying and cosmetic type.

Author Response
Dear Editor and Reviewer,
We would like to thank you for the time allowed to this review process. As a result, we are submitting the revised version for a possible publication in this respectable Journal. Below, you can find our responses; each comment is followed by its respective reply. We made changes in the manuscript in order to address suggestions and make it clearer for the readers, we underlined in yellow the responses to your comments and we used the track changes to correct some misprint or to enhance some phrases of the manuscript. All authors have made sufficient contributions and have approved the submitted manuscript.
Sincerely,
The Authors
Legend:
R2 (Reviewer 2)
A (Authors)
1) R2:
I was asked to review the paper 3D Kinematic Analysis of Walking and Nordic Walking with 2 Different Poles’ Length. The work is very nicely written. I have quite a few comments, but they are of the clarifying and cosmetic type.
A:
Thank you very much, we are glad you appreciated the paper. Thanks for you comments.
2) R2:
I think this title would be more relevant: The impact of Nordik Walking pole length on gait kinematic parameters.
Line 3 - Do not insert a period after the title. Please remove it.
A:
We changed the title as suggested. We will inform even the Editor and Reviewer 1 about this change just to involve also them in the final choice.
3) R2:
Lines 24 – 26 – The sentence is too long, making it grammatically incorrect. Please divide it into two sentences.
A:
Done. Line 27.
4) R2:
Lines 30 – 32 – The results should be written more precisely. I don't know what angular displacement is. I mean that not all parameters are defined. They are clear only after the explanations in Table 4.
A:
Done. To avoid long sentence and due to the impossibility to go in deep with details in the abstract, we tried to use a more generic term that can help the reader. Lines 32-33.
5) R2:
Lines 56 – 57 – please expand on the sentence “controversial results”.
A:
Done. Lines 58-59.
6) R2:
Lines 71-72 – Is the person's height in centimeters or meters, please add this information.
A:
Done. Line 74.
7) R2:
Lines 75 – 76 – please add information about the weight range of the poles.
A:
Done. Lines 78-79.
8) R2:
Line 84 – Please remove the hypothesis, as it is obvious. If you do not want to remove, please specify whether it will change the kinematic parameters of the lower limbs or only the upper limbs?
A:
Thank you for the comment. We prefer to maintain the hypothesis, therefore we add a specification as requested. We hope it can be appreciated. In case it does not, we will remove the hypothesis as suggested. Lines 88-91.
9) R2:
Line 97 – I don't understand what it means that the persons were not experts. Was there an assumption that these people had never practiced NW? Or did they have contact with it but rare. If rare, what does that mean?
A:
Never practiced. We correct it. Line 102.
10) R2:
Line 103 – please describe this technique in a separate subsection. Not everyone is an expert in NW. To me, it is not obvious. Was there a specified time of day for training and terrain?
A:
Thanks for your comment, this will enhance the quality of the paper. We have added the required information. Lines 109-112.
11) R2:
Line 105 – Were there women and men among the participants?
A:
No, only men. Sorry it was a typo. We add “male” in line 113. Probably it was canceled in our draft revision because in the abstract and discussion (limitation) we explicated that we had a male sample. Thank you for this comment. Line 113.
12) R2:
Section Instrumentation – please add a photo of the person at the measuring station.
A:
Done. Figure 1. Lines 136-140.
13) R2:
Line 147 – please put the period before Table 1.
A:
Done. Line 161.
14) R2:
Lines 160 – 162 – cosmetic change. I suggest that you write instead of dashes – hyphens as follows: W – walking; …..
A:
Done. Lines 174-176.
15) R2:
Lines 169 – 170 – That is, how many gait cycles were taken? Why did you decide to analyze the right side?
A:
We took approximatively 60 cycles and the right side was chosen for two reasons: 1) it was the dominant side for the whole sample; 2) moreover it was the side with fewer interruptions in the kinematic signal therefore less interpolation of the data was required.
16) R2:
Section 2.3.1 – I propose to precede the text with tables and figure. That is, immediately after the text, where the table is mentioned it should appear. The figure may be reduced in size. In this form, the text is read badly. In addition, Figure 2 and Table 2 can be combined in a drawing as follows:
A:
Thanks for the advice. Done. Lines 183-187.
17) R2:
Table 3 – C7 vertical displacement – is this range of motion? If yes, please change the name?
A:
No, it isn’t. Figure A1 in the Appendix can help to understand. It is the linear vertical displacement from the lower to the higher points. It is an average value of all cycles.
18) R2:
Are the parameters in Table 3 average values? If so, you need to add where are the average and where are the maximum values and from how many cycles (I already mentioned this issue).
A:
We understand the point and we agree. We have added a note at the bottom of the table. Lines 201-202.
19) R2:
Table 4 – SL/EHD ratio parameter (cm) appeared in the results. It should be discussed in methods. Why it is used.
A:
It is already described in methods: lines 192-195.
20) R2:
Figure 3 – please increase the font size as it is not readable.
A:
Done.